# OpenReview forum: "Failures to Find Transferable Image Jailbreaks Between Vision-Language Models"
_ICLR.cc/2025/Conference — ICLR 2025 Poster_

### Official Review · Reviewer_WSM9 · 2024-10-22

**Soundness:** 4
**Presentation:** 4
**Contribution:** 4
**Rating:** 8
**Confidence:** 4

**Summary:**

The authors study the transferability of adversarial attacks between Vision Language Models (VLM). Through a comprehensive experimental work, the authors show that adversarial attacks show limited transferability between VLM(s). This is unlike transferability on image classifiers and Language Models where attacks are more likely to transfer. As a result, this work indicates that VLM(s) are more robust to adversarial attacks. It also prompts the question: are there more transferable attacks against VLM(s)?

**Strengths:**

- The writing and presentation are clear.
- The experimental work is comprehensive and impressive.
- The problem is well motivated.
- The conclusions are interesting and quite impactful.
- Good care was put into accurately measuring the success of the attack.

**Weaknesses:**

- In the setting where attacks partially transfer if models were identically initialized (3.3), how important is the identical initialization? It is unclear from reading the paper. Would the transferability break if the models were initialized differently?
- Can the authors provide any intuition on why they think there was no transfer to the 2 stage model in Section 3.6 even when the ensembles are scaled?
- Can the authors clarify what they mean by a jailbreak image? Is it a randomly initialized image that is then optimized adversarially? Moreover, is it one image for all prompts? If that is the case, could the authors comment if more images could improve the success of the attack, perhaps grouped per prompt topic? Furthermore, how about optimizing attacks based on relevant images to the topic rather than images that are pure noise? I don’t think additional experiments are required here, just clarifications that could be incorporated into future work.

**Questions:**

Refer to the weaknesses section. Overall, the work is experimentally strong and of practical importance. Therefore, I recommend acceptance.

---

> ### Author Response · Authors · 2024-11-16
> **Response to Reviewer WSM9**
>
> Thank you for reviewing our paper. We are grateful you found the problem well motivated, the writing clear, the experiments comprehensive and well motivated and our conclusions impactful.
>
> For calibration purposes, we’d like to note that the ICLR 2025 rubric differs slightly from previous similar conferences. For example:
>
> - To indicate "Accept", the NeurIPS 2024 rubric says to use 7 whereas the ICLR 2025 rubric says to use 8
> - To indicate "Strong Accept", the NeurIPS 2024 rubric says to use 9 whereas the ICLR 2025 rubric says to use 10
>
> To address the concerns you raised:
>
> > In the setting where attacks partially transfer if models were identically initialized (3.3), how important is the identical initialization? It is unclear from reading the paper. Would the transferability break if the models were initialized differently?
>
> > Can the authors provide any intuition on why they think there was no transfer to the 2 stage model in Section 3.6 even when the ensembles are scaled?
>
> These two questions are actually related! Consider two VLMs that are initialized with the same vision backbone and same language models and same MLP, but one of the VLMs is trained in a 2-stage process where in the first stage, only the MLP is trained, and then in the second stage, both the MLP and the language model are trained. We can think of this 2-stage training as equivalent to initializing the two VLMs’ MLPs differently and then doing 1-stage training. In our experiments, this 2-stage training blocks successful transfer (to be clear, this may not be a general phenomenon since different attacks could yield potentially different results).
>
> Our hypothesis for why different MLP initialization blocks transfer is a combination of two factors: (1) that there are many many many ways to transform the outputs of the vision model into inputs to the language model, and thus using differently initialized MLPs causes the VLM training to move the same initial language model in different ways, and (2) the image jailbreak optimization we are using is too unconstrained such that the attack rapidly succeeds against the attacked VLM(s) but is unable to transfer to new target VLMs.
>
> We will add language in the discussion section to address your questions explicitly in the paper..
>
> > Can the authors clarify what they mean by a jailbreak image? Is it a randomly initialized image that is then optimized adversarially? Moreover, is it one image for all prompts? [...] I don’t think additional experiments are required here, just clarifications that could be incorporated into future work.
>
> Yes, that’s correct. A good reference is “Universal and Transferable Adversarial Attacks on Aligned Language Models” https://arxiv.org/abs/2307.15043. The goal is to obtain a single input image that “unlocks” all the bad behavior of a target VLM that is otherwise suppressed due to safety and alignment training. We will clarify this in our Discussion and advocate that future work should explore the experiments you suggest.

---

> > ### Comment · Reviewer_WSM9 · 2024-11-27
> > **Reply**
> >
> > Thanks for clarifying my concerns! My positive score remains unchanged.

---

### Official Review · Reviewer_rWpX · 2024-11-02

**Soundness:** 2
**Presentation:** 2
**Contribution:** 2
**Rating:** 3
**Confidence:** 4

**Summary:**

The paper analyses the transferability between VLM jailbreaks. In particular, previous work has demonstrated that image-level jailbreaks do not transfer between different VLMs. The paper analyses this finding on a suite of 49 prismatic and various other VLMS. Overall, they find that image-level jailbreaks do mostly not transfer between different VLMs and transfer is only possible if the models are very similar to each other (jailbreak created on an ensemble of very similar VLMs to the target model with the same architecture and training data).

**Strengths:**

- The paper's topic is relevant, as transferable attacks would enable black-box attacks on VLMs and make it easy to target many such models. Additionally, if obfuscation of model weights is a way to safe against image-level jailbreaks, this could make open-source models less safe than close-weights models.
- The main message of the paper—that VLM image jailbreaks do not transfer well—is clearly conveyed, and the empirical evidence suggests this isn’t simply due to a lack of effort. This finding also seems to be validated by earlier results from Bailey et al. (2023), Qi et al. (2024a), and Chen et al. (2024b).
- I appreciate the structure of the paper, especially the breakdown of transferability across different granularities. It effectively demonstrates that there is some transfer between very similar models but that this drops off surprisingly quickly as models become less similar (e.g., in training data, training scheme, or architecture). This is especially surprising given how well adversarial image attacks typically transfer across different image classifiers.
- The evaluation pipeline and experimental setup appear reasonable. The choice of using prismatic VLMs also facilitates the exploration of transferability, and I appreciate the inclusion of newer VLMs like Qwen.

**Weaknesses:**

- Attack Methodology: The authors rely on a single attack using 50k steps with Adam, initializing from either random noise or a single image. I think 50k steps is excessive; it might make sense to check once if an increased number of steps helps, but from Figures 4, 5, and 6, there doesn’t seem to be any justification for going beyond 5k steps (except for a few control trials to demonstrate this). This is relevant since the authors state that "Due to computational limitations, we were unable to explore more sophisticated attacks". It just seems unreasonable to explore 50k steps for every attack instead of trying out a larger variety of optimizers, step sizes, weight decays, and so on. I believe that this is essential.
- Schlarmann et al. (2023) used APGD, a well-regarded method for evaluating adversarial robustness. Unlike the attack used in this paper, APGD is usually used with an epsilon constraint. Would it be possible to try out APGD to see if these attacks transfer? I believe it would be the strong baseline. It might also be interesting to ablate the attack radius epsilon in this context. Also, this paper uses 5k steps as well so it would be a lot faster to evaluate than the author's 50k attack.
- I  don’t mind publishing negative results, and the literature generally supports the findings here (with Niu et al. (2024) as a notable exception). However, if the paper’s main contribution is to show that transferable jailbreaks are unachievable, it needs a more robust approach than a single attack. I consider this a must-have for this paper. This also relates to the 2 previous points.
- The authors start from a single image and use this to claim that starting from a natural image is no different than starting from pure noise
First, various datasets available online could serve as initialization sources, and starting from only one image isn’t enough to claim that noise and image initialization are equivalent.

**Questions:**

- Are the image pixel values clipped to [0,1] during both optimization and evaluation? Additionally, does the VLM see a uint8 representation during the final evaluation, or is the optimized example used directly in floating-point precision without discretization? If so, does this impact the results?

---

> ### Author Response · Authors · 2024-11-16
> **Response to Reviewer rWpX**
>
> Thank you for reviewing our paper. We are grateful you found our topic relevant, our message clearly conveyed and our experimental methodology reasonable.
>
> For calibration purposes, we’d like to note that the ICLR 2025 rubric differs slightly from previous similar conferences. For example:
>
> - To indicate "Accept", the NeurIPS 2024 rubric says to use 7 whereas the ICLR 2025 rubric says to use 8
> - To indicate "Strong Accept", the NeurIPS 2024 rubric says to use 9 whereas the ICLR 2025 rubric says to use 10
>
> To address the concerns you raised:
>
> > I think 50k steps is excessive; it might make sense to check once if an increased number of steps helps, but from Figures 4, 5, and 6, there doesn’t seem to be any justification for going beyond 5k steps (except for a few control trials to demonstrate this).
>
> Respectfully, we disagree. While our results _now_ demonstrate that going beyond 5k steps make little difference, at the time, such a result was not obvious. Results in ML such as grokking suggest that surprising phenomena can occur after optimizing for much longer. We didn’t mention this in the paper, but when working on the project, we actually first ran our experiments with 5k steps, then realized that we couldn’t draw conclusive results, so we reran the experiments for 50k steps.
>
> >  It just seems unreasonable to explore 50k steps for every attack instead of trying out a larger variety of optimizers, step sizes, weight decays, and so on. I believe that this is essential.
>
> > However, if the paper’s main contribution is to show that transferable jailbreaks are unachievable, it needs a more robust approach than a single attack. I consider this a must-have for this paper.
>
> > Schlarmann et al. (2023) used APGD, a well-regarded method for evaluating adversarial robustness.
>
> We do agree that exploring different optimization methods is a clear way we can improve our paper. We will run what we can during the review process and post updates as we obtain them, starting with your suggested APGD. What prioritized sequence of experiments do you think would most improve the work? Would you like us to try other optimization methods afterwards, or should we try different image initializations?
>
> > Are the image pixel values clipped to [0,1] during both optimization and evaluation?
>
> Yes, that’s correct.
>
> > Does the VLM see a uint8 representation during the final evaluation, or is the optimized example used directly in floating-point precision without discretization? If so, does this impact the results?
>
> The image is optimized in fp16, saved as uint8, and then loaded and evaluated in fp16. We tested whether that loss of precision in saving the image makes a difference and found no significant effect. We could also more rigorously test this and include such results in the paper, if you would like. What, in your opinion, would constitute convincing evidence?

---

> > ### Comment · Reviewer_rWpX · 2024-11-17
> >
> > > Respectfully, we disagree. While our results now demonstrate that going beyond 5k steps make little difference, at the time, such a result was not obvious. Results in ML such as grokking suggest that surprising phenomena can occur after optimizing for much longer. We didn’t mention this in the paper, but when working on the project, we actually first ran our experiments with 5k steps, then realized that we couldn’t draw conclusive results, so we reran the experiments for 50k steps.
> >
> > As I said, I also think that it is important to show that you don't get non-transferable attacks with 5k steps but at 50k suddenly attacks become transferrable. I am also not against running everything with 50k steps, I just think that having more varied attacks is very important, and that might be cheaper to do with fewer steps.
> >
> > > We do agree that exploring different optimization methods is a clear way we can improve our paper. We will run what we can during the review process and post updates as we obtain them, starting with your suggested APGD. What prioritized sequence of experiments do you think would most improve the work? Would you like us to try other optimization methods afterwards, or should we try different image initializations?
> >
> > I think running vanilla APGD from the initialization you used with a standard attack radius should be the first priority. And again, I think you can show on selected experiments that the number of steps above 5K does not change results and then just run a larger evaluation with fewer steps during the rebuttal period.
> >
> > As for initialization, I think it might make sense to see if images that are more related to the question perform differently than images that are not related. For example, you used the "how does one make anthrax" question in your Figure. Now, does my transfer rate change if I start with 10 images that are related to anthrax or other chemical weapons compared to starting with 10 images of cute puppies? Ideally, you'd have a few different object categories with about 10 images, and show average transfer rates across these categories and see if this influences results.
> >
> > > The image is optimized in fp16, saved as uint8, and then loaded and evaluated in fp16. We tested whether that loss of precision in saving the image makes a difference and found no significant effect. We could also more rigorously test this and include such results in the paper, if you would like. What, in your opinion, would constitute convincing evidence?
> >
> > I agree that this is the correct way to do this. I think you could consider an experiment where you measure transfer rates between the images in floating point precision and their uint8 variants, though I do not think that this is super important. And obviously this also only makes sense if all preprocessing is implemented for both float and uin8 images and there's no conversion in the preprocessing stage. But for me, this is not super important, it would just be great to have this mentioned somewhere in the appendix.

---

> ### Author Response · Authors · 2024-11-20
> **Quick Question about Auto-PGD for Reviewer rWpX**
>
> Reviewer rWpX, you recommended we include other attacks, beginning with APGD. Recalling that our goal is to find _transferable_ image jailbreaks, do you know how successful APGD is as a transfer attack? If APGD is not good at producing transferable attacks, could you please recommend other baselines?

---

> ### Author Response · Authors · 2024-11-21
> **Response to Reviewer rWpX**
>
> To circle back, we agree that running additional attacks will strengthen the paper and we have such runs underway. In the interim, we want to sync with you (Reviewer rWpX) to clarify two points.
>
> > I think running vanilla APGD from the initialization you used with a standard attack radius should be the first priority.
>
> Regarding APGD, we are unsure why APGD makes sense. APGD was designed to be better at successfully attacking models, and to the best of our knowledge, was not designed to maximize the chance of successful transfer. For comparison, we find that we can easily optimize universal image jailbreaks against the attack VLM(s), whereas our goal is to find robust transfer between VLM(s). Why is APGD sensible in our setting? If it isn't, could you please suggest other attacks that are aimed at obtaining successful transfer?
>
> > As for initialization, I think it might make sense to see if images that are more related to the question perform differently than images that are not related. For example, you used the "how does one make anthrax" question in your Figure. Now, does my transfer rate change if I start with 10 images that are related to anthrax or other chemical weapons compared to starting with 10 images of cute puppies?
>
> The experiments we currently have running use arbitrary natural images for initialization. Regarding your suggestion, we are unsure whether it makes sense for our manuscript.
>
> 1. Choosing images that are intentionally paired with the prompt contradicts with our goal of obtaining _universal_ jailbreaks. Our motivation is to find GCG-like images that elicit harmful outputs for _any_ prompt. The suggested experiment would deviate from this goal of obtaining transferable and universal image jailbreaks.
>
> 2. The suggestion has already been done (in various forms and flavors) by at least four different papers: (i) Shayegani et al.'s, "Jailbreak in pieces: Compositional Adversarial Attacks on Multi-Modal Language Models", (ii) Gong et al.'s "FigStep: Jailbreaking Large Vision-language Models via Typographic Visual Prompts", (iii) Li et al.'s "Images are Achilles’ Heel of Alignment: Exploiting Visual Vulnerabilities for Jailbreaking Multimodal Large Language Models" and (iv) Liu et al.'s "MM-SafetyBench: A Benchmark for Safety, Evaluation of Multimodal Large Language Models". The general consensus is that showing a VLM an image makes the VLM more willing to talk about the image, although success varies depending on the particular model.
>
> 3. We think the experimental methodology you suggested needs more care. Comparing ASRs conditioned on anthrax pictures vs conditioned on puppy pictures introduces a confounder of how willing a VLM is to generate text related to that input image. Rather, we think the correct comparison is whether an anthrax+optimized image is more successful against other VLMs than the anthrax image alone. Would you agree?

---

### Official Review · Reviewer_SUQ4 · 2024-11-03

**Soundness:** 3
**Presentation:** 3
**Contribution:** 3
**Rating:** 6
**Confidence:** 5

**Summary:**

This paper mainly investigates the transferability issue of image adversarial attacks among Visual Language Models (VLMs). The research findings show that the image adversarial attacks optimized for single or multiple VLMs are generally difficult to transfer to other VLMs that have not been attacked. Only when the attacking model is highly similar to the target model can partial successful transfer be observed.

This paper proposes a method to optimize image adversarial attacks by attacking a collection of multiple VLMs that are highly similar to the target VLM. The experimental results demonstrate that this method can significantly improve the success rate of attacks on the target VLM, making it close to complete success.

**Strengths:**

1. This paper conducts a large-scale empirical study on more than 40 different types of VLMs, including different vision backbones, language models, VLM training data, and optimization strategies, systematically investigating the transferability issue of image adversarial attacks. This investigative study provides materials for subsequent research work.
2. The research results show that it is generally difficult for general-purpose image adversarial attacks to transfer between VLMs. Only when the attacking model is highly similar to the target model can partial successful transfer be observed. This is a key observation results in research communities.
3. To improve transferability, this paper proposes a method to optimize image adversarial attacks by attacking a collection of multiple VLMs that are highly similar to the target VLM. The experimental results show that this method can significantly improve the success rate of attacks on the target VLM.

**Weaknesses:**

1. The definition of the "highly similar" VLM is not clear. Although the author briefly described what a highly similar VLM is in the paper, it doesn't necessarily mean that the models are highly similar. There are a lot of research works that are related to evaluate the similarity of NN, as [1-5] shown. Also some of them study the similarity of NN via the loss landscapes, which is similar to some in this paper.
I suggest that the author refer to these works to investigate the actual similarity of the models. And further explain the capabilities of transfer attacks in combination with quantitative analysis.
Thus, my major concerns can be detailed as follows: 1) Provide a quantitative measure of similarity between VLMs, perhaps using techniques from the papers in [1-5]. 2) Analyze how this quantitative similarity measure correlates with the success of transfer attacks. 3) Does the current research contribute to the analysis of the similarity of VLMs?

2. According to Figure 9, it seems that only one photo was used when initializing with natural images. This may lead to biases in the test results. Although the author also conducted some experiments on initializing with random noise, it is also very necessary to carry out evaluations with more initializations using natural images.
So, please conduct more experiments with multiple diverse natural image initializations. Analyze whether the choice of natural image initialization affects the transferability of attacks. If there are differences based on initialization, discuss the implications for real-world attacks.


[1] Similarity of Neural Architectures using Adversarial Attack Transferability. ECCV 2024.

[2] Similarity of neural network representations revisited. ICML 2019.

[3] Sharp minima can generalize for deep nets. ICML 2017.

[4] Visualizing the loss landscape of neural nets. NeurIPS 2018.

[5] How do vision transformers work? ICLR 2022.

**Questions:**

Please refer to the weaknesses.

---

> ### Author Response · Authors · 2024-11-16
> **Response to Reviewer SUQ4**
>
> Thank you for reviewing our paper. We are glad that you think it provides useful material for subsequent work.
>
> For calibration purposes, we’d like to note that the ICLR 2025 rubric differs slightly from previous similar conferences. For example:
>
> - To indicate "Accept", the NeurIPS 2024 rubric says to use 7 whereas the ICLR 2025 rubric says to use 8
> - To indicate "Strong Accept", the NeurIPS 2024 rubric says to use 9 whereas the ICLR 2025 rubric says to use 10
>
> To address the concerns you raised:
>
> > Thus, my major concerns can be detailed as follows: 1) Provide a quantitative measure of similarity between VLMs, perhaps using techniques from the papers in [1-5]. 2) Analyze how this quantitative similarity measure correlates with the success of transfer attacks. 3) Does the current research contribute to the analysis of the similarity of VLMs?
>
> We agree that this would be a novel and useful contribution, but such work would likely constitute 1-2 standalone research papers to identify different notions of similarity for VLMs (which, to the best of our knowledge, is an open question) and study how different notions of similarity for VLMs connect to task-relevant quantities, of which adversarial robustness is only one. Such a contribution would be valuable, but would require significant additional research.
>
> Our paper aims to test whether image jailbreaks can successfully transfer to new VLMs, and if so, under what conditions. We believe our claim that close model similarity increases transferability to be supported by our existing experiments - specifically, in Figure 6 we see a direct correlation between the amount of additional training given to the transfer model and lower transfer. We believe that this experiment, combined with our partial transfer results in Figure 4, adequately support the impact of model similarity, despite lacking a quantitative measure of similarity. We consider identifying and evaluating different measures of similarity between VLMs to be beyond the scope of this work.
>
> > The definition of the "highly similar" VLM is not clear. Although the author briefly described what a highly similar VLM is in the paper
>
> We’re afraid that the terminology of “highly similar” might have detracted from the point we aimed to communicate. If we could work together to improve how the manuscript communicates, that would be much appreciated.
>
> To expound, we don’t have a precise quantitative notion of what it means for two VLMs to be similar. We even explicitly stated that “we lack a mathematical definition of ‘highly similar’.” Our goal was to find transferable image jailbreaks against VLMs, and we were able to do so by attacking ensembles of VLMs comprised of (i) the same starting image backbone as the target + (ii) the same starting language model as the target + (iii) the same starting MLP as the target + (iv) similar VLM training recipes as the target + (v) similar VLM training data as the target. Colloquially, these attacked VLMs are highly similar to the target, but we lack a quantitative metric of “similarity” and we need a compact way to refer to this combination of VLM properties for exposition.
>
> What terminology would you advise? How can we better communicate this point while (i) remaining succinct and (ii) remaining agnostic to the notion of similarity between VLMs?
>
> > To improve transferability, this paper proposes a method to optimize image adversarial attacks by attacking a collection of multiple VLMs that are highly similar to the target VLM. The experimental results show that this method can significantly improve the success rate of attacks on the target VLM.
>
> If we can clarify this point, we don’t view attacking ensembles of VLMs that are highly similar to the target as a practical attack. Rather, we wanted to show that we were able to achieve transfer, albeit under highly unrealistic conditions. We want future work to focus on figuring out what conditions of either the VLM or the attack are sufficient to obtain strong transfer against black-box VLMs under realistic conditions. We hope our demonstration of transfer in this limited case can serve as a starting point for future work to further develop transferability. We will clarify this in the manuscript.
>
> > According to Figure 9, it seems that only one photo was used when initializing with natural images [...] Analyze whether the choice of natural image initialization affects the transferability of attacks.
>
> We will run these experiments and add the results to the manuscript within the next week. We will update you once the results have been added.

---

### Official Review · Reviewer_yyLr · 2024-11-04

**Soundness:** 3
**Presentation:** 3
**Contribution:** 3
**Rating:** 8
**Confidence:** 4

**Summary:**

The paper investigates the transferability of gradient-based image jailbreaks across various vision large language models (VLLMs). Through comprehensive empirical analysis involving 40 diverse VLM architectures, the study aims to understand if adversarial images that prompt harmful responses in one or several models simultaneously can induce similar outputs in others. The adversarial images, jointly optimized on several VLLMs over a set of harmful requests and responses, should show some transferability following existing  work. However, this paper shows that even under some ideal setting, transfer of the adversarial images  is quite difficult.

**Strengths:**

1. The studied problem is of central interest to the community and the results are count-intuitive and interesting.

2. The paper conducts extensive experiments and considers several many reasonable settings, making the conclusion convincing.

3. The paper is well-written and easy to follow. It clearly states its conclusions without making overblown claims.

**Weaknesses:**

1. The main conclusion of this paper is that image jailbreaks do not successfully transfer between VLMs. It is unclear whether this issue arises from limited transferability or from the content itself being harmful (thus triggering the safety alignment mechanisms of VLLMs). I would be interested in seeing the transfer results for non-harmful content. For example, VLLMs consistently generating the same text, regardless of the input text.

2. Missing discussion of overfitting. Regarding the ASR on the optimized dataset as the train metric and the transfer ASR as the test metric, finding a transferable image jailbreak is basically a generalization problem.
 For the text-only LLM attack, the optimization space is $V^N$ where $N$ is the length of the adversarial string and $V$ is the vocabulary size. Typically $N=20, V\in[32000, 160000]$, making $\sim10^{100}$ possibilities. For VLLM attack, the space is much large: $255^{HW}$ where $HW$ is the resolution of the input image, making $\sim10^{1000000}$ possibilities..  The likelihood of overfitting increases exponentially for the problem studied in this paper. One evidence is that the timescale to jailbreak does not change pretty much in different settings since   the optimization difficulties are the same for different settings. It is possible that 8 ensemble models are enough for text-only transfer but way far from enough for image transfer.

(This is not a weakness, but rather some suggestions that may enhance the paper.) The generalization gap can be more effectively measured using probably approximately correct (PAC) learning theories. Existing research on transfer attacks for image classifiers and image MLLMs, some of which has already been cited, also demonstrates that strategies to reduce overfitting can improve transferability.

**Questions:**

1. Since the input texts in the dataset don’t refer to any images (only some harmful requests), will the attention mechanism simply ignore the input image and reduce transferability?

2. Does the order of the image and text input matter the transferability? GCG inserts the adversarial suffix after the original prompt, but some of the VLLMs listed in the paper insert the image before the text prompt.

3. Will the optimized jailbreak images follow some similar pattens?

---

> ### Author Response · Authors · 2024-11-16
> **Response to Reviewer yyLr**
>
> Thank you for reviewing our paper. We are grateful you found our work of central interest to the community and that our results are counter-intuitive and interesting!
>
> For calibration purposes, we’d like to note that the ICLR 2025 rubric differs slightly from previous similar conferences. For example:
>
> - To indicate "Accept", the NeurIPS 2024 rubric says to use 7 whereas the ICLR 2025 rubric says to use 8
> - To indicate "Strong Accept", the NeurIPS 2024 rubric says to use 9 whereas the ICLR 2025 rubric says to use 10
>
> To address the concerns you raised:
>
> > Missing discussion of overfitting. [...] For the text-only LLM attack, the optimization space is […] The likelihood of overfitting increases exponentially for the problem studied in this paper [...] It is possible that 8 ensemble models are enough for text-only transfer but way far from enough for image transfer.
>
> This is a great point to integrate into our Discussion. To help us shape how we communicate this point, we’re unsure whether generalization is the correct term for transferability since generalization typically refers to how the test loss differs from the training loss, whereas here, the object that changes isn’t the data fed into the model but rather which model we feed the data into. What is the correct terminology for this?
>
> We agree with the intuition that as the degrees of freedom significantly increase, the probability of obtaining robust transferability without constraints or regularization probably tends towards zero. Do you know of any prior work that we should appropriately cite?
>
> In this new discussion paragraph, we will state that one future direction is identifying what constraints or regularization or other algorithmic changes enable more robust transfer between VLMs.
>
> > The main conclusion of this paper is that image jailbreaks do not successfully transfer between VLMs. It is unclear whether this issue arises from limited transferability or from the content itself being harmful (thus triggering the safety alignment mechanisms of VLLMs). I would be interested in seeing the transfer results for non-harmful content. For example, VLLMs consistently generating the same text, regardless of the input text.
>
> This seems reasonable. What would be a good way to quantitatively assess this? We would be happy to run experiments if you can guide us on what experiments you would find compelling. Specifically:
> - What tasks should we focus on?
> - For those tasks, which datasets would be appropriate? Should we focus on text-only datasets or image+text datasets or both, since we are studying VLMs?
> - How should we measure the similarity of generated text? For instance, if I ask “What object is in this photo?” and one VLM answers “Dog” while another answers “Golden Retriever”, how do we quantify whether the “same” text has been generated?
> - How should we ensure we sample a reasonable diversity of inputs?
>
> > Since the input texts in the dataset don’t refer to any images (only some harmful requests), will the attention mechanism simply ignore the input image and reduce transferability?
>
> We’re not sure that this specific hypothesis can explain the evidence. Specifically, the optimized image jailbreak is as irrelevant to the harmful prompts for the attacked VLM(s) as for the transferred VLMs, but the optimized image successfully jailbreaks the attacked VLM(s) but fails to transfer. If attention mechanisms generically do not attend to irrelevant input images, we should expect any irrelevant image to fail to jailbreak even attacked VLMs, which contradicts what we observe. We think your hypothesis of overfitting to the attacked VLM(s) is the likely explanation. If you disagree with this reasoning, please let us know!
>
> > Does the order of the image and text input matter the transferability? GCG inserts the adversarial suffix after the original prompt, but some of the VLLMs listed in the paper insert the image before the text prompt.
>
> We took the VLMs as given. We felt that experimenting with VLM architectures would add significant complexity to our experiments, as well as potentially decrease the applicability of our results to the field since such VLMs would be non-standard.
>
> > Will the optimized jailbreak images follow some similar pattens?
>
> The only noticeable pattern we observed was that the borders of each of the patches of the optimized image darkened. This makes the images look like they’ve been overlaid with a grid-like pattern.

---

### Author Response · Authors · 2024-11-21
**Strong Supporting Evidence by Recent Work**

Dear reviewers,

We would like to draw your attention to recent work https://arxiv.org/abs/2410.03489, which studies gradient-based image jailbreaks in _fusion_-based vision-language models (VLMs). This work is complementary to ours, as we study _adapter_-based vision-language models (VLMs).

This new work finds results similar to ours: they are able successfully optimize images to jailbreak attacked VLMs, but the image jailbreaks do not transfer to new VLMs (Section 6). The new work also uses a different attack than we do.

We view this work as offering additional support that VLMs seem to exhibit robustness to transfer attacks in a manner that eludes language models and image classifiers. It's quite possible that different attacks will yield transferable image jailbreaks, and future work should try to identify the necessary and/or sufficient conditions to create such transferable image jailbreaks.

---

### Meta-Review · Area_Chair_dWQo · 2024-12-21

**Metareview:**

This paper focuses on the transferability of image adversarial jailbreaks across Visual Language Models (VLMs). The findings reveal that image adversarial attacks optimized for one or multiple VLMs generally struggle to transfer effectively to other, unoptimized VLMs. Partial transfer success is observed only when the attacking model closely resembles the target model. To address this limitation, the paper introduces a method that optimizes image adversarial attacks by targeting a group of VLMs with high similarity to the target model. Experimental results show that this approach significantly enhances the attack's success rate on the target VLM, achieving near-complete success. Reviewers agree the paper has the following advantages: (1) The studied problem is important and the results are count-intuitive and interesting. (2) The paper conducts extensive experiments and considers several reasonable settings, making the conclusion convincing. (3) The paper is well-written and easy to follow. One reviewer is still concerned about the problem of using a single image for experiments and the conclusions only validate in the specified setups. Considering the comments and discussion of other reviewers, I agree the concerns did not harm the main contributions but the final version should address the concerns with more discussion and experiments reported in the rebuttal.

**Additional Comments On Reviewer Discussion:**

All reviewers provided solid comments in the first round of reviewing and discussed with the authors during the rebuttal process. Three reviewers agreed the paper could be accepted, while one reviewer still concerned with the experiment setups of the paper during the discussion and gave a low rate (Reject). I carefully check the comments of the reviewer and the opinions of other reviewers. The concerns did not harm the key contributions and conclusions.

---

### Decision · Program_Chairs · 2025-01-22

Accept (Poster)